# Detection of COVID-19 Patients Using Machine Learning Techniques: A Nationwide Chilean Study

**DOI:** 10.3390/ijerph19138058

**Published:** 2022-06-30

**Authors:** Pablo Ormeño, Gastón Márquez, Camilo Guerrero-Nancuante, Carla Taramasco

**Affiliations:** 1Escuela de Ingenieria y Negocios, Universidad de Viña del Mar, Viña del Mar 2520000, Chile; 2Departamento de Electrónica e Informática, Universidad Técnica Federico Santa María, Millennium Nucleus on Sociomedicine, Concepción 4030000, Chile; 3Escuela de Enfermería, Universidad de Valparaíso, Valparaíso 2500000, Chile; 4Facultad de Ingeniería, Universidad Andrés Bello, Millennium Nucleus on Sociomedicine, Viña del Mar 2520000, Chile

**Keywords:** Epivigila, machine learning, symptoms, comorbidities

## Abstract

Epivigila is a Chilean integrated epidemiological surveillance system with more than 17,000,000 Chilean patient records, making it an essential and unique source of information for the quantitative and qualitative analysis of the COVID-19 pandemic in Chile. Nevertheless, given the extensive volume of data controlled by Epivigila, it is difficult for health professionals to classify vast volumes of data to determine which symptoms and comorbidities are related to infected patients. This paper aims to compare machine learning techniques (such as support-vector machine, decision tree and random forest techniques) to determine whether a patient has COVID-19 or not based on the symptoms and comorbidities reported by Epivigila. From the group of patients with COVID-19, we selected a sample of 10% confirmed patients to execute and evaluate the techniques. We used precision, recall, accuracy, $F_{1}$-score, and AUC to compare the techniques. The results suggest that the support-vector machine performs better than decision tree and random forest regarding the recall, accuracy, $F_{1}$-score, and AUC. Machine learning techniques help process and classify large volumes of data more efficiently and effectively, speeding up healthcare decision making.

## 1. Introduction

Humanity, throughout its history, has been threatened by several epidemiological events, such as smallpox, cholera, Zika virus and, currently, the COVID-19 pandemic. In these situations, epidemiological surveillance is an essential public health action, which allows collecting, analysing, and interpreting health data to describe and monitor a health event to support the planning, implementation, and evaluation of public health interventions and programmes [1].

Epidemiological surveillance aims at the early detection and timely management of cases of people requiring health services and also aims to establish a diagnosis of the health status of the population as a whole in order to manage disease prevention and control measures at the population level [2]. An important challenge for epidemiological surveillance systems arises in epidemics, endemics and pandemics, where diseases are required to be reported immediately so that control and follow-up measures can be generated [3].

Since the beginning of the pandemic, Chilean authorities have been working to coordinate and establish guidelines for the benefit of public health in order to implement the TTI (testing, traceability and isolation) strategy. In this regard, the Chilean Ministry of Health has instructed all clinicians to report suspected cases and active searches through a platform called Epivigila [4]. This platform is a virtual monitoring system that allows the tracking of COVID-19 and has become the most important strategic tool for the Chilean government to support public health decision making. The system allows the notifier, in this case, a clinician, to register patient and contact information, which will be sent to primary care centres to begin traceability. Subsequently, COVID-19 follow-up delegates can initiate the Chilean protocol to follow up on detected cases.

Epivigila has more than 60,000 users, more than 6,000,000 patient records under epidemiological surveillance and 17,000,000 accumulated notifications, making it an essential source of data for studies related to the detection and dissemination of COVID-19 in the Chilean population. Moreover, the Chilean Ministry of Health uses Epivigila as a source of information to calculate the health indicators reported to the Chilean population daily [4]. Given the vast volume of data handled by Epivigila, manual analysis of the data conducted by the stakeholders related to the Chilean surveillance process is insufficient to make the best of its information source regarding classifying COVID-19 infected patients. Both symptoms and comorbidities are essential attributes for the quantitative and qualitative analysis of the cumulative prevalence rate in Chile, i.e., the total number of new diagnosed cases (confirmed and probable).

In order to process considerable volumes of data efficiently, effectively and in the shortest possible time, machine learning techniques, such as support-vector machines (SVM), random forests (RF) and decision trees (DT), are indispensable to identify patterns and classify data. SVM is concerned with the correlation of data represented in a high-dimensional feature space. In this space, data points are categorized, even if it is not possible to separate them linearly [5]. RF is a machine learning technique that has the ability to be generalized to myriad classification problems [6]. DT is a supervised learning technique mainly used in problems related to data classification. In this regard, it is possible to use DT for both categorical and continuous variables [7].

This paper compares the SVM, DT and RF techniques to determine whether a patient has COVID-19 or not using real-time data based on symptoms and comorbidities managed by Epivigila. To evaluate the performance of the techniques, we used a sample of 10% COVID-19 patients out of the total of confirmed cases reported in Epivigla. We used the metrics precision, recall, accuracy, harmonic mean between precision and recall ($F_{1}$-score) and the area under the curve (AUC). The main contribution of our paper is that it is an analytical study comparing machine learning techniques on a dataset to distinguish whether a patient has COVID-19 or not with data obtained in real-time and generated from clinical processes led by clinicians in Chile.

## 2. Related Work

Ahamad et al. [8] developed a machine learning methodology whose primary focus was to systematically identify the most relevant clinical symptoms to predict true positive COVID-19 cases. The authors of this study validated the predictions using COVID-19 patient data from seven provinces in China. The study results demonstrated that contact with infected individuals is an adequate predictor, but this depends on a rigorous contact tracing process and social network analysis. Validation indicated that muscle pain and diarrhea are significant symptoms of COVID-19.

Aydin et al. [9] studied the factors affecting the number of positive cases and deaths associated with the COVID-19 pandemic using a new model proposed by the study authors, which consisted of three steps. First, countries were grouped according to the data provided. Second, the effectiveness of the clusters was assessed using WSIDEA (weighted stochastic imprecise data envelopment analysis). Third, the performances of the algorithms were compared in terms of success criteria.

Awal et al. [10] proposed a model to detect COVID-19 which was focused on machine learning. The main feature of this model is the ability to detect COVID-19 in a short time. The study used different classification techniques, such as linear discriminant analysis (LDA), quadratic-DA (QDA), naive Bayes (NB), k-nearest neighbors (KNN), decision tree (DT), random forest (RF), estreme gradient boosting (XGB), gradient boosting (GB), and support-vector machine (SVM), among others, for the rapid detection of COVID-19 through the symptoms of patients and the estimation of their status (infected or not infected). Analyzing these techniques helps physicians in detecting COVID-19 from the clinical data of hospitalized patients as well as taking preventive measures while treating patients with COVID-19.

Monaghan et al. [11] used a machine learning model that addresses the risk of a hemodialysis patient with an undetected COVID-19 infection. The authors used extreme gradient boosting (XGBoost), a scalable and distributed gradient-boosted decision tree machine learning technique. The study usds 40,490 hemodialysis patients to build the model (11,166 COVID-19-positive cases and 29,324 unaffected patients). The results of the study described that the machine learning model proposed by the authors was reasonable for predicting hemodialysis patients at risk of COVID-19 within three days before there is clinical suspicion.

Shaban et al. [12] addressed the problem of classifying COVID-19 patient diagnoses by assigning a weight to each feature in the classification model. The authors’ technique was based on three sequential phases, which were: (i) the pre-processing phase, (ii) the feature classification phase, and (iii) the classification phase. In turn, the authors identified several input features obtained from laboratory findings.

The studies mentioned in this section describe promising results regarding using machine learning techniques to predict infected patients and other conditions related to COVID-19. Our study contributes to the body of knowledge by comparing different machine learning techniques to determine whether a patient has COVID-19 or not using real-time, volatile, and unstructured data from the Chilean population.

## 3. Materials and Methods

The methodology used in our study is described in Figure 1. First, we explored the Epivigila system and the Chilean epidemiological surveillance process. Then, we defined the steps to clean and format the Epivigila data. Subsequently, we trained and tested the machine learning techniques. Finally, we evaluated the performance of the techniques and analyzed the corresponding results.

### 3.1. Data Source

In 2019, a technological integration platform for notifiable diseases called Epivigilia began to be used by the Chilean health network. This system supports the implementation of public health strategies for the notification, monitoring and control of the COVID-19 pandemic in Chile. Its versatility positions this platform among the few globally that operate national data in a pandemic with a high degree of granularity integrated into a single system [4].

In Chile, Epivigila is one of the main and most important sources of information for daily reports, epidemiological reports and governmental websites [4]. The main objective of Epivigila is to improve the quality of information for timely decision-making and execution of the necessary actions for the protection of the health of the population and to ensure compliance with information security standards. Epivigila strengthens surveillance processes through real-time monitoring of the occurrence of diseases with epidemic potential through strong articulation with the actors of the surveillance network. Additionally, it contributes to improving the management of outbreaks and epidemics once they are detected.

The notification process of Epivigila starts when the patient consults a clinician (public or private). When the clinician detects a suspected COVID-19 case, he or she must log in to Epivigilia and notify the case (see Figure 2).

The data are automatically routed to the Provincial Health Services, where they are validated, and then continue to the Regional Health Services to certify the cases reported. Subsequently, the data are forwarded to the Chilean Ministry of Health authorities to prepare daily epidemiological reports. In addition to the reporting of cases, the system also receives the notification of the reverse transcription polymerase chain reaction (RT-PCR) tests, which health authorities consider to be the only valid test for diagnosing COVID-19.

The epidemiological surveillance process allows the clinical characteristics of patients affected with COVID-19 to be known, which means that risk groups can be identified at an early stage. Indeed, studies such as [13,14] describe common characteristic symptoms and comorbidities in patients infected or who have died of COVID-19-related causes. However, when the amount of data to be processed is enormous and cannot be efficiently analysed by health professionals, the ability to identify risk groups early on is lost, limiting public health decision-making.

### 3.2. Data Pre-Processing

The Epivigila system has 1939 fields containing sensitive patient data, symptoms, diseases, and geographic locations, among other things. The total of patient records handled by Epivigila is 6,000,000. From this number, 600,000 (10%) correspond to confirmed cases and 5,400,000 (90%) correspond to suspects or discarded cases. The confirmed cases correspond to those patients tested for COVID-19 and confirmed as having COVID-19. The suspected patients are those who presented symptoms but were not confirmed or discarded. Finally, the discarded cases are those patients who tested negative for COVID-19. Through random sampling of the second group (suspects or discarded), we selected 600,000 cases in order to balance the classes. We separated the total cases (1,200,000) into 80% for training (960,000) and 20% for testing (240,000). Additionally, we selected 60,000 confirmed cases for validation (control group).

Given the vast volume of data, in this article, we propose an initial study focusing on symptoms and comorbidities using a reduced number of fields of the Epivigila system in order to evaluate the results of SVM, DT and RF in an optimal time frame. The original dataset contains different levels of suspicion with respect to COVID-19. For this study, we only consider the confirmed and rejected cases to perform a binary analysis of these classes. After identifying the cases, we proceed to clean the data to identify missing values. In this regard, we find two different types of data: (i) patients who do not report symptoms and (ii) comorbidities of patients who do not report symptoms. For the latter case, instead of eliminating these data, we have considered the value “no report” in order to know whether these data are relevant or not.

Consequently, in order to pre-process the symptom and comorbidity data, we used a binary vector that identifies the occurrence of the symptom and comorbidity with “1” and “0”, in case of absence. We model the vector as “1” if the symptom or comorbidity appears for the patient and “0” if it does not. Therefore, if the patient does not report a health problem that the health professional did not detect, it is marked as “no report” and has a “0” in our model. Additionally, we consider in the same vector the variable “none” when a patient has no symptoms. Table 1 describes the symptoms and comorbidities used in our study.

### 3.3. Learning Phase

As we depicted in Figure 1, a training set is used to train three machine learning algorithms, described as follows.

#### 3.3.1. Support-Vector Machine (SVM)

A support-vector machine is one of the most popular supervised machine learning techniques which is focused on regression and classification tasks. The goal of SVM is to discover a hyperplane in *N*-dimensional space (*N*—the number of salient points) that particularly classifies information foci. Since SVM satisfies several classification properties, this technique is generally used in different types of research [5].

In our study, we use SVM type *C* [15]. This type of SVM minimizes the objective function for a C>0. This value is called the adjustment constant and is expressed by the following equation:(1)$$\tau \left(w,\xi \right)=\frac{1}{2}{\left|w\right|}^{2}+C\overset{m}{\underset{i_{1}}{\sum \xi_{i}}}$$
where *C* is the penalty parameter of error that the degree of correct classification model can meet; *w* is the normal vector to the hyperplane; and $\xi_{i}$ is the support-vector error threshold variable. The constant *C* determines the trade-off between maximizing the margin and minimizing the training error.

#### 3.3.2. Decision Tree (DT)

A decision tree is a classification technique aimed at optimizing both classification and regression of data [7]. DT uses a tree representation in which each leaf corresponds to a group of elements and a branch corresponds to a value recursively. DT uses the concept of entropy, which is defined by a variable *Y* whose potential values have probabilities $p_{1},p_{2},...,p_{k}$. The estimation of *Y* is described as follows:(2)$$Entropy\left(Y\right)=-\sum_{j}p_{j}log_{2}\left(p_{j}\right)$$

The goal of a DT is to build a tree for all data and process an individual output on each leaf. The statistical property (better known as information gain) can determine how well an attribute separates a set of information based on the target classification. An attribute in a node with high information gain can organize the data better to improve classification accuracy. The following formula allows us to obtain the information gain IG of an attribute *X* in relation to a training data set *D* (where *E* is entropy):(3)$$IG\left(D,X\right)=E\left(D\right)-\sum_{v\in Values(X)}\frac{D_{v}}{D}E\left(D_{v}\right)$$

In this equation, the set of values of attributes *X* is defined as Values(X), and $D_{v}$ is the subset of *D* for which attribute *X* has a value *v*. The information gain is calculated for all attributes of a node in a tree. The attribute with the most significant information gain is selected as the best attribute that splits the data correctly.

#### 3.3.3. Random Forest (RF)

Random forest corresponds to a set of regression and classification trees. RF trains training datasets called bootstraps and can combine them to obtain a more accurate result. Bootstraps are created from a random sampling of the training dataset [6]. RF can work with large, higher-dimensional data sets with comparatively higher accuracy. In our study, we consider the following training set, ${\left\{\left(x_{i},y_{i}\right)\right\}}_{i=1}^{n}$, as input. Then, we initialize model with a constant value:(4)$$F_{0}={\mathrm{argmin}}_{\gamma }\sum_{j=1}^{n}L\left(y_{j},\gamma \right)$$
where $F_{0}$ is a constant value to initialize the model, and L(yF(x)) a differentiable loss function. Subsequently, for m=1,…,M, we compute the pseudo-residuals as follows:(5)$$r_{im}={\left[\frac{\delta L(y_{i},F\left(x_{i}\right)}{\delta L(F\left(x_{i}\right)}\right]}_{F\left(x\right)-F_{m-1}\left(x\right)},for\;i=1,\ldots ,n$$

Then, we fit the base learner (tree) $h_{m}\left(x\right)$ to pseudo-residuals, i.e, train it using training set ${\left\{\left(x_{i},r_{im}\right)\right\}}_{i=1}^{n}$. We compute multipliers by solving the one-dimensional optimization problem as follows:(6)$$\gamma_{m}={\mathrm{argmin}}_{\gamma }\sum_{j=1}^{n}L\left(y_{j},F_{m-1}\left(x_{i}\right)+\gamma h_{m}\left(x\right)\right)$$

Finally, we update the model:(7)$$F_{m}=F_{m-1}\left(x_{i}\right)+\gamma h_{m}\left(x\right))$$

### 3.4. Performance Analysis

Each technique was trained on 80% of the available data and tested on 20%. To tune parameters, these were adjusted using 5-group cross-validation. For SVM, we used a linear kernel and C=1. We used 250 estimators for RF, and for DT, no tuning was used.

To distinguish between an actual class and the predicted class, we used the labels $\left\{Y,N\right\}$. Given a classifier and an instance, we have four possible outputs. If the instance is positive and the classification is positive, we obtain a true positive (TP); if it is classified as negative, we obtain a false negative (FN). In turn, if the instance is negative and is classified as negative, we obtain a true negative (TN); if it is classified as positive, we obtain a false positive (FP) [16].

The true positive rate (tpRate) is defined as follows:(8)$$tpRate=\frac{TP}{p}=\frac{TP}{TP+FN}$$

On the other hand, the false positive rate (fpRate) is defined as follows:(9)$$fpRate=\frac{FP}{N}=\frac{FP}{FP+TN}$$

Accordingly, we define the following metrics to assess the performance of SVM, DT, and RF.
(10)$$precision=\frac{TP}{TP+FP}$$
(11)$$recall=\frac{TP}{TP+FN}$$
(12)$$accuracy=\frac{TP+TN}{P+N}$$
(13)$$F_{1}-score=\frac{2TP}{2TP+FP+FN}$$
(14)$$Specifity=\frac{TN}{TN+FP}$$

Additionally, we use the area under the curve (AUC) [17] to organise our classifier and visualise its performance. This curve is commonly used in medical decision making and to evaluate the performance of machine learning and data mining.

We divided our dataset into four criteria (age 0–20, 21–60, 61–96 and 0–96). Subsequently, we separated our dataset into two parts: one part (70%) for training and another part (30%) for testing.

## 4. Results

### 4.1. Demographics

Regarding the range of 0 to 96 years of male subjects, the average age is 39 years and the interquartile range (IQR) is 27 to 53 for 3,533,360 (52.1%). Similarly, regarding the range of 0 to 96 years of female subjects, the average age is 38 years old and the IQR is 27 to 54 for 3,229,717 (47,9%).Table 2 shows the association of patient COVID-19 confirmations and some selected demographic information. Of those studied patients, 2,443,628 (36.0%) patients expressed symptoms, and 3,471,634 (51.1%) patients expressed comorbidities, whereas among confirmed patients, 79.3% developed symptoms. On the other hand, 19.7% of patients had headaches, which was the most frequent symptom, and 43.7% presented with high blood pressure, which was the most frequent comorbidity; their body temperature was equal to 36.5 centigrade degrees.

Figure 3 depicts the age-wise total number of patients. In the age range of 25 years to 65 years, the rate of individuals affected is higher than in children and other adults.

Figure 4 indicates the frequency of each symptom for all the patients, and Figure 5 describes the frequency for all the comorbidities.

Figure 6 and Figure 7 depict the percentage relative frequency of symptoms and comorbidities of confirmed and discarded patients.

### 4.2. Machine Learning Analysis

We identified the five most significant features (shown in Table 3) that are strictly related to COVID-19-positive status. In our analysis results we found that every algorithm achieved an accuracy score of 70% or above. The performances of our used algorithms for the different datasets are described in Table 4.

Table 5 depicts the accuracy measurement methods and their score for the age range of (i) 0 to 20 years, (ii) 21 to 60 years, (iii) 61 to 96 years and (iv) 0 to 96 years.

Regarding the range of 0 to 20 years, the precision for SVM is 0.613, which is the lowest. RF and DT have an identical score, which is 0.628. On the other hand, SVM provides the highest recall value with a score of 0.68, and the scores for RF and DT are 0.604 and 0.558, respectively. Concerning the $F_{1}$-score, the highest value corresponds to SVM and is 0.645. Concerning AUC, SVM obtained the highest value. We observed that abdominal pain, dyspnoea and obesity were the most significant features in this age range.

Regarding the range of 21 to 60 years, the precision for SVM is 0.717, which is the lowest. RF achieves a score of 0.735, and DT achieves a score of 0.731. SVM provides the highest recall value with a score of 0.785, and the scores for RF and DT are 0.728 and 0.697, respectively. Regarding the $F_{1}$-score, the highest value corresponds to SVM and is 0.749. Concerning AUC, SVM obtained the highest value. We observed that chronic lung disease, myalgia, and odynophagia were the most significant features in this age range.

Concerning the range of 61 to 96 years, the precision for SVM is 0.730, which is the highest. RF and DT score 0.717 and 0.718, respectively. SVM provides the highest recall value with a score of 0.811, and the scores for RF and decision tree are 0.690 and 0.607, respectively The highest value of $F_{1}$-score corresponds to SVM and is 0.768. Concerning AUC, SVM obtained the highest value. We also observed that fever, cough, and asthma were the more significant features of this age range.

Finally, regarding the range of 0 to 96 years, the precision for SVM is 0.727, which is the lowest. RF achieves a score of 0.739, and DT achieves a score of 0.746, which is the highest. SVM provides the highest recall value with 0.798, and the scores for RF and DT are 0.740 and 0.684, respectively. Additionally, the $F_{1}$-score for SVM is 0.761, which is the highest. Concerning AUC, SVM obtained the highest value. We observed that loss of taste, myalgia, and dyspnoea were the most significant features of this age range.

## 5. Discussion

The use of representative data from the universe of positive cases of COVID-19 in Chile has generated consistent scientific evidence, and this is of high value with respect to improving public health measures, such as protection strategies for susceptible populations and proposals for targeted vaccinations [18,19].

According to the results obtained in our study, SVM was the best performing technique in the detection of COVID-19 patients. SVM stood out from the other techniques because of the *C* parameter of the SVM, which indicates the degree of misclassification of each training set to be avoided. For large values of *C*, the optimization chooses a smaller margin hyperplane. which implies that the hyperplane does a better job of correctly classifying all training points. On the other hand, a small *C* value causes the optimizer to choose a large margin separator hyperplane, even if that hyperplane misclassifies many points.

Machine learning allows analyses to be established with a high volume of data, as observed in our study. The results obtained in our research can be the baseline for a predictive analysis of mortality in the Chilean population. This is consistent with the findings of Mahdavi et al., who used the support-vector machine technique to estimate mortality predictions [20]. In particular, the studies of different algorithms have compared each one according to the data obtained. In general, our results showed that random forest showed regular performance compared to SVM and DF. In other studies, RF has shown the best performance in a study with smaller data volumes than those obtained in our research [21].

This study gives results with a group of data representative of the national context. According to Mondal et al. in a review study, they concluded that the use of ML with symptomatology data and diagnostic tests is a beneficial application to diagnose patients infected with COVID-19 [22]. This same study indicated that it is relevant to conduct artificial intelligence studies with larger data sets [22].

The findings of the clinical manifestations concerning symptoms and comorbidities were consistent with the scientific evidence developed in countries such as Brazil, which shared a similar circulation of COVID-19 variants at the time of our study [23].

Concerning the limitations of our study, Epivigila contains more than 17,000,000 records that are updated daily. The number of COVID-19-positive cases in the system corresponds to more than 700,000 patients. Since processing this amount of data requires a more sophisticated technological infrastructure, this study uses a sample of 170,000 COVID-19-positive patients. We selected this number of patients because this number is the maximum amount of data that our current technological infrastructure can process. Although we cannot generalize the performance results of the machine learning techniques with this amount of data, the results obtained are promising for preliminary analyses of the symptoms and comorbidities of COVID-19-positive patients in order to support the quantitative analysis of the Chilean Ministry of Health.

In summary, our results have shown that machine learning can be useful in the field of epidemiology and public health. Having this type of tool can also facilitate, in the clinical field, potential applications for diagnosis or estimating the magnitude of the problem through the risk of mortality or forecasting infections. In fact, scientific evidence has shown its usefulness in diagnostic tools using imaging data [24], epidemiological forecasts [25] and the transfer of scientific knowledge to the practices of health workers [26].

## 6. Conclusions

This article presents a study about the analysis of three machine learning techniques, support-vector machine (SVM), decision tree (DT), and random forest (RF), to determine whether a patient has COVID-19 or not based on symptoms and comorbidities managed by the Chilean epidemiological surveillance system Epivigila. From a total of more than 600,000 COVID-19-positive patients reported by the Epivigila system, we selected a sample of patients to test the performance of the techniques. The results obtained by our study allow us to verify that SVM performed better than RF and DT for detecting infected patients using a large volume of data. Due to the hyperplane selection properties of SVM, the training in which we chose all points to classify symptoms and comorbidities is optimal compared to RF and DT.

Our future work will focus on developing and deploying a more robust technological infrastructure to process the full range of COVID-19-positive patients. Additionally, we are defining a work plan that consists of proposing algorithms and techniques for both machine learning and big data to process Epivigila COVID-19 patient data in order to forecast and evaluate new symptoms and comorbidities to support healthcare decision-making in Chile.

## Figures and Tables

**Figure 1 ijerph-19-08058-f001:**
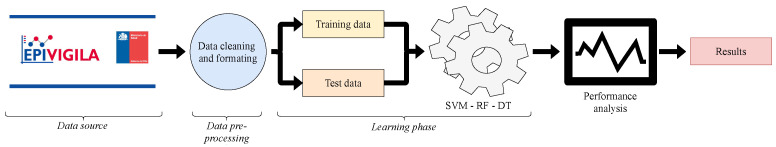
Proposed methodology.

**Figure 2 ijerph-19-08058-f002:**
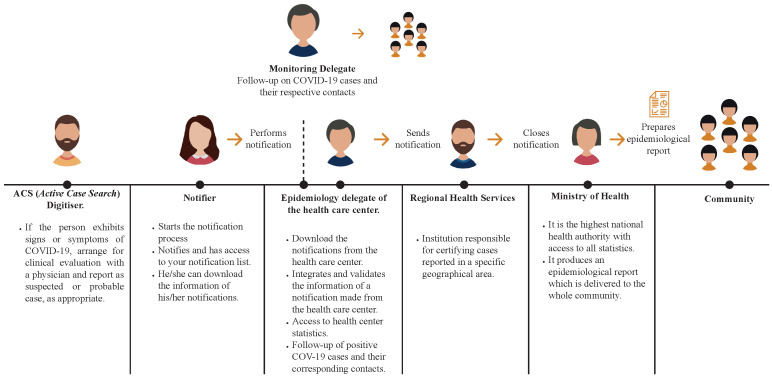
Epidemiological surveillance process conducted in Chile.

**Figure 3 ijerph-19-08058-f003:**
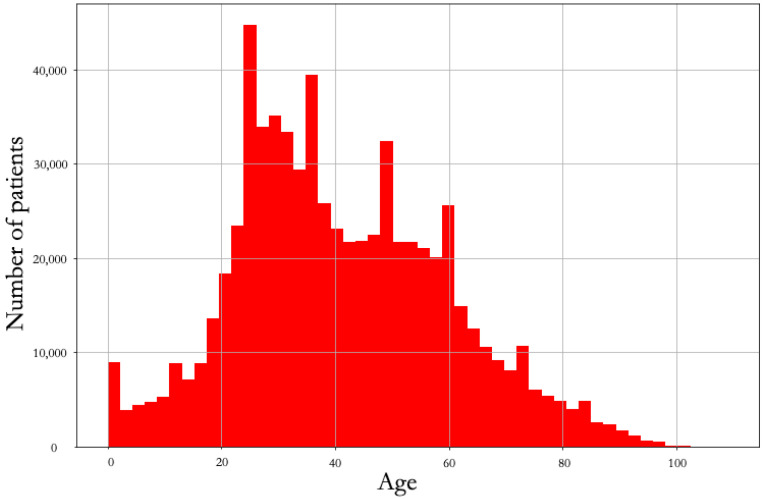
Distribution of confirmed patients by age.

**Figure 4 ijerph-19-08058-f004:**
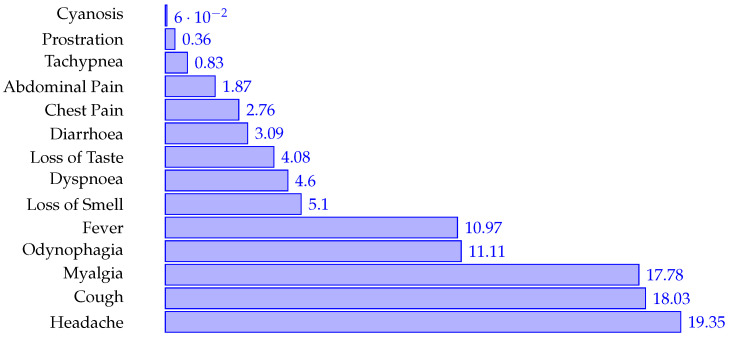
Symptoms frequency.

**Figure 5 ijerph-19-08058-f005:**
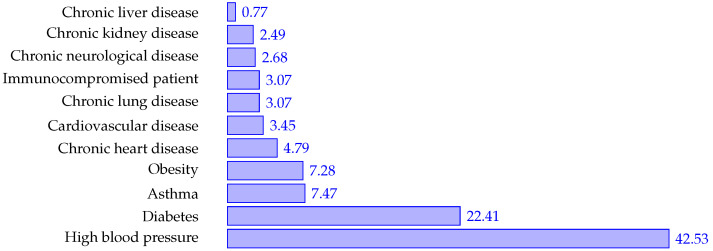
Comorbilities frequency.

**Figure 6 ijerph-19-08058-f006:**
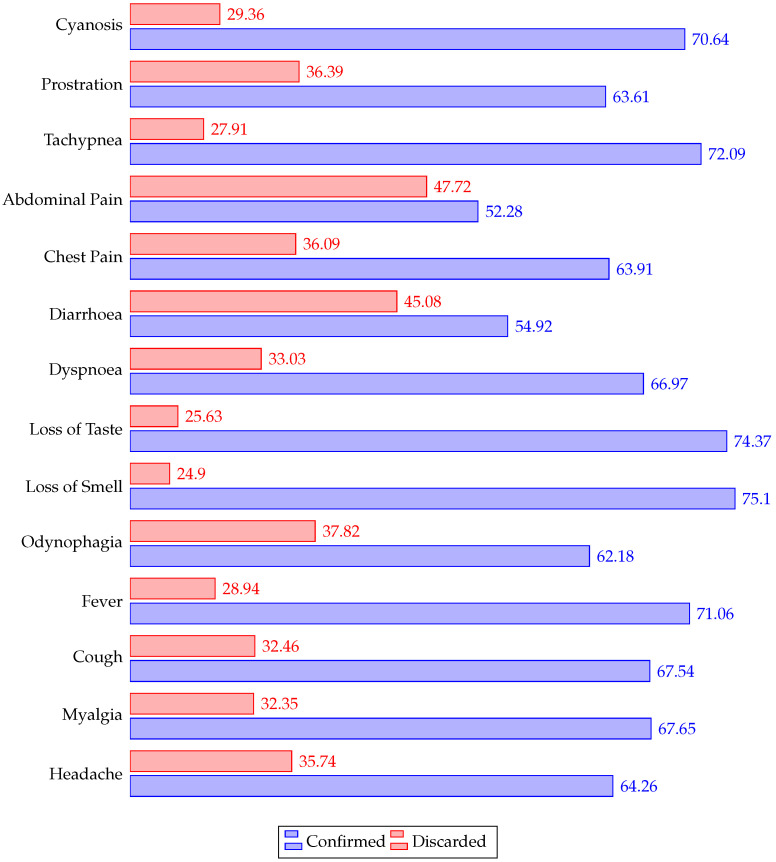
Relative frequency (percentage) of symptoms in confirmed and discarded patients.

**Figure 7 ijerph-19-08058-f007:**
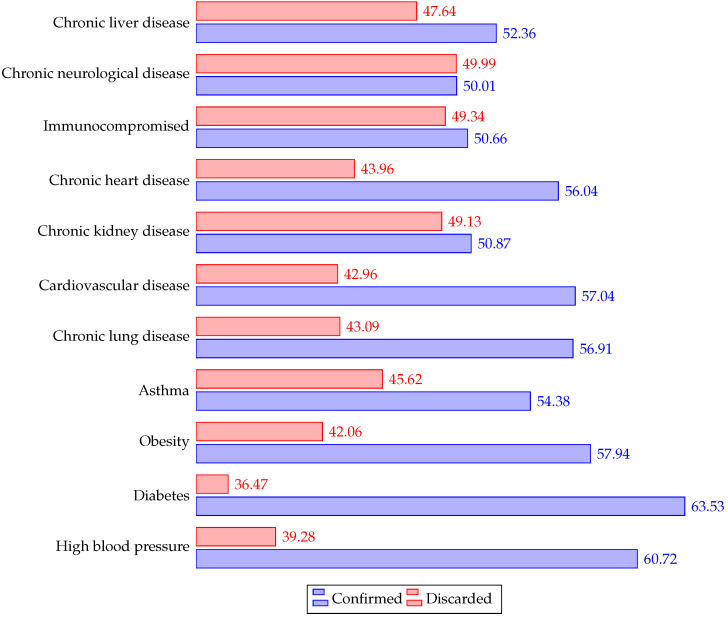
Relative frequency (percentage) of comorbilities in confirmed and discarded patients.

**Table 1 ijerph-19-08058-t001:** Symptoms and comorbidities used in our study.

Symptoms	Comorbidities
Tachypnoea	Asthma
Odynophagia	Chronic kidney disease
Cyanosis	Chronic lung disease
Abdominal pain	High blood pressure
Headache	Obesity
Fever	Immunocompromised patient
Diarrhoea	Chronic heart disease
Loss of taste	Diabetes
Myalgia	Chronic neurological disease
Chest pain	Chronic liver disease
Prostration	Cardiovascular disease
Dyspnoea	
Cough	
Loss of smell	

**Table 2 ijerph-19-08058-t002:** Male and female demographics.

All Patients	Suspected	Confirmed	Total
Mean age (interquartile range)	37 (27–52)	36 (26–51)	39 (28–54)
Male gender (%)	52.1%	52.7%	51.2%
Female gender (%)	47.9%	47.3%	48.8%
Have symptoms (%)	52.6%	33.3%	79.3%

**Table 3 ijerph-19-08058-t003:** Top 5 features reported by SVM, RF and DT techniques categorized by age ranges.

Dataset	Technique	Top 5 Features				
		1st	2nd	3rd	4th	5th
Age (0–20)	SVM	Abdominal pain	Loss of taste	Chronic kidney disease	Tachypnea	Chronic liver disease
	RF	Chronic heart disease	Odynophagia	Diarrhoea	Cough	Fever
	DT	Abdominal pain	Odynophagia	Diarrhoea	Loss of smell	Cough
Age (21–60)	SVM	Abdominal pain	High blood pressure	Chronic kidney disease	Asthma	Diabetes
	RF	Abdominal pain	Diarrhoea	Chronic heart disease	Cough	Fever
	DT	Abdominal pain	Cough	Odynophagia	Dyspnoea	Chronic heart disease
Age (61–96)	SVM	Abdominal pain	Diabetes	Loss of taste	Odynophagia	Fever
	RF	Abdominal pain	Cough	Diarrhoea	Chronic heart disease	Fever
	DT	Abdominal pain	Cough	Diarrhoea	Chronic heart disease	Fever
Age (0–96)	SVM	Abdominal pain	Chronic lung disease	Immunocompromised patient	Diabetes	Loss of taste
	RF	Abdominal pain	Cough	Diarrhoea	Chronic heart disease	Fever
	DT	Abdominal pain	Chronic heart disease	Cough	Cardiovascular disease	Odynophagia

**Table 4 ijerph-19-08058-t004:** Precision, recall, $F_{1}$-score, specificity, and AUC results for SVM, RF and DT techniques categorized by age ranges. Intense red indicates the highest values.

Dataset	Technique	Precision	Recall	$F_{1}$-Score	Specificity	AUC
	SVM	0.613	0.680	0.645	0.574	0.640
Age (0–20)	RF	0.628	0.604	0.616	0.712	0.636
	DT	0.628	0.558	0.591	0.737	0.626
Age (21–60)	SVM	0.717	0.785	0.749	0.705	0.739
	RF	0.735	0.721	0.728	0.758	0.732
	DT	0.731	0.667	0.697	0.792	0.712
Age (61–96)	SVM	0.730	0.811	0.768	0.687	0.753
	RF	0.717	0.690	0.704	0.747	0.705
	DT	0.718	0.607	0.658	0.779	0.680
Age (0–96)	SVM	0.727	0.798	0.761	0.681	0.748
	RF	0.739	0.740	0.740	0.740	0.738
	DT	0.746	0.684	0.713	0.784	0.724

**Table 5 ijerph-19-08058-t005:** SVM, RF and DT accuracy results categorized by age range. Intense red indicates the highest values.

	0–20 Years			21–60 Years			61–96 Years			0–96 Years		
	SVM	RF	DT	SVM	RF	DT	SVM	RF	DT	SVM	RF	DT
Abdominal Pain	0.9712	0.0711	0.1946	0.9962	0.1973	0.5565	0.9930	0.1723	0.4304	0.9919	0.1859	0.4294
Asthma	0.0016	0.0346	0.0527	0.0002	0.0211	0.0216	0.0000	0.0421	0.0359	0.0000	0.0053	0.0055
Cardiovascular disease	0.0000	0.0458	0.0379	0.0000	0.0289	0.0197	0.0000	0.0160	0.0231	0.0000	0.0308	0.0395
Chest pain	0.0001	0.0363	0.0484	0.0000	0.0042	0.0039	0.0000	0.0132	0.0169	0.0000	0.0312	0.0187
Chronic heart disease	0.0001	0.0946	0.0276	0.0000	0.1043	0.0292	0.0000	0.0676	0.0331	0.0000	0.0813	0.0657
Chronic kidney disease	0.1198	0.0052	0.0046	0.0632	0.0036	0.0043	0.0000	0.0137	0.0125	0.0000	0.0111	0.0141
Chronic liver disease	0.0849	0.0028	0.0049	0.0000	0.0152	0.0166	0.0000	0.0134	0.0104	0.0000	0.0039	0.0045
Chronic lung disease	0.0003	0.0116	0.0093	0.0000	0.0126	0.0170	0.0000	0.0224	0.0076	0.0001	0.0288	0.0247
Chronic neurological disease	0.0000	0.0029	0.0014	0.0000	0.0106	0.0094	0.0000	0.0088	0.0068	0.0000	0.0027	0.0003
Cough	0.0000	0.0793	0.0617	0.0000	0.0886	0.0334	0.0000	0.1032	0.0283	0.0000	0.0879	0.0566
Cyanosis	0.0002	0.0639	0.0516	0.0000	0.0308	0.0130	0.0000	0.0304	0.0302	0.0000	0.0326	0.0102
Diabetes	0.0007	0.0275	0.0367	0.0000	0.0193	0.0203	0.0000	0.0169	0.0170	0.0000	0.0148	0.0157
Diarrhoea	0.0000	0.0820	0.0680	0.0000	0.1221	0.0168	0.0000	0.0770	0.0432	0.0000	0.0863	0.0303
Dyspnoea	0.0001	0.0390	0.0394	0.0000	0.0268	0.0297	0.0000	0.0518	0.0465	0.0000	0.0403	0.0333
Fever	0.0000	0.0734	0.0464	0.0000	0.0744	0.0212	0.0000	0.0606	0.0093	0.0000	0.0652	0.0107
Headache	0.0283	0.0010	0.0022	0.0000	0.0001	0.0000	0.0000	0.0035	0.0026	0.0000	0.0014	0.0036
High blood pressure	0.0001	0.0035	0.0016	0.1019	0.0008	0.0012	0.0000	0.0152	0.0164	0.0000	0.0060	0.0054
Immunocompromised patient	0.0001	0.0010	0.0018	0.0000	0.0012	0.0020	0.0000	0.0173	0.0173	0.0000	0.0275	0.0299
Loss of smell	0.0001	0.0625	0.0645	0.0000	0.0329	0.0174	0.0000	0.0305	0.0268	0.0000	0.0375	0.0270
Loss of taste	0.4791	0.0521	0.0490	0.0000	0.0249	0.0268	0.0000	0.0228	0.0193	0.0000	0.0259	0.0281
Myalgia	0.0000	0.0487	0.0405	0.0000	0.0485	0.0257	0.0000	0.0158	0.0192	0.0000	0.0435	0.0253
Obesity	0.0000	0.0016	0.0013	0.0000	0.0030	0.0030	0.0000	0.0351	0.0168	0.0000	0.0193	0.0170
Odynophagia	0.0000	0.0820	0.0802	0.0000	0.0576	0.0316	0.0000	0.0277	0.0318	0.0000	0.0590	0.0380
Prostration	0.0000	0.0043	0.0056	0.0000	0.0013	0.0007	0.0000	0.0087	0.0129	0.0000	0.0078	0.0063
Tachypnea	0.1129	0.0061	0.0105	0.0000	0.0245	0.0233	0.0000	0.0376	0.0174	0.0000	0.0089	0.0047

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
