# Peer review of "Detection of COVID-19 Patients Using Machine Learning Techniques: A Nationwide Chilean Study"

_ijerph, 2022, doi:10.3390/ijerph19138058_

Round 1

Reviewer 1 Report

The authors aimed to predict and classify symptoms, comorbidities, and other conditions related to COVID-19, which is not a common practice in machine learning modeling. Usually, the symptoms and comorbidities would be utilized to develop models to predict an outcome status. However, the authors use the same terminology throughout the manuscript, which might mislead the readers to interpret their results falsely. 

Abstract: the "cases" should be specified.

Why selected only 10% of confirmed cases?

SVM performed better by how much difference? Need to give a detailed quantitative description of the comparison 

Features used in the models should be reported.

Line 79: WSIDEA should be spelled out. The algorithms should be specified. 

Line 81: the performance of the models should be reported. In addition, the rationale of the current study reported in this manuscript should be given. In other words, why do we need other prediction models? Additionally, how their models help physicians take preventive measures while treating patients with 84 COVID. COVID should be replaced with COVID-19. 

Line 88: models predicting hemodialysis is out of the scope of this manuscript and should be removed. 

line 155: "in order to know whether these data are 156 relevant or not" is not clear

Parameters in formulas should be specified. 

Table 2: p-values were wrong. 

4.1. The authors should clearly report in an algorithm and numbers and include patients in their final dataset. 

The authors should not only report frequency among positive cases (seemingly Figure 4 and 5) but also the relative frequency (RR or OR) among positive and negative cases. The feature importance ranking should also be provided to interpret the models. 

It is not clear how the authors tune their models. They should specify whether they used the testing dataset to fine-tune their models. 

Author Response

Comment 1: The authors aimed to predict and classify symptoms, comorbidities, and other conditions related to COVID-19, which is not a common practice in machine learning modeling. Usually, the symptoms and comorbidities would be utilized to develop models to predict an outcome status. However, the authors use the same terminology throughout the manuscript, which might mislead the readers to interpret their results falsely.

Answer: The objective of this work is not to classify symptoms or comorbidities. We use these features as input of our model, along with the patient's age, to determine if the patient can be confirmed with COVID-19 or not. In particular, we use it as a feature vector with 0 o 1 depending on the existence or not of the symptom or the comorbidity. Therefore, we have improved the message and title of the manuscript to make it clear that we use symptoms and comorbidities as characteristics to determine confirmed or unconfirmed cases of COVID-19.

Comment 2: Abstract: the "cases" should be specified.

Answer: Thank you very much for your comment. We agree with the reviewer that the word "cases" is ambiguous. Therefore, we rewrote (line 8) the sentence where we specified that the cases are COVID-positive patients.

Comment 3: Why selected only 10% of confirmed cases?

Answer: Thank you very much for your comment. In the original dataset, the confirmed cases are only the 10\% of the total. The other classes (90\%) correspond to discard cases or suspected cases, so we tried to maintain the proportion in each sample.

Comment 4: SVM performed better by how much difference? Need to give a detailed quantitative description of the comparison

Answer: Thank you very much for the comment. Undoubtedly, the reviewer addresses an interesting aspect of our article. In our study, we have used performance metrics that are defined in equations 10, 11, 12, 13 and 14. Based on these metrics, in section 4.2 we discuss the results of each machine learning technique. In this regard, we realized that SVM had the best recall in all age ranges. Although the literature, in general, suggests that better precision determines the performance of a machine learning technique, this judgment depends on the context. Given that our study analyzes the predictions of the outputs of machine learning techniques, the recall should be high since we are also interested in false negatives. Having said this, SVM presented a high recall in all age ranges, so we consider that SVM had a better performance than the rest of the techniques.

Comment 5: Features used in the models should be reported.

Answer: Thank you very much for your comment. For every model, the features used were the 14 symptoms and the 11 comorbidities (see Table 1). Additionally, the age of the patient was used. Finally, the classes were divided into confirmed and discarded.

Comment 6: Line 79: WSIDEA should be spelled out. The algorithms should be specified.

Answer: Thank you very much for your comment. We have added the WSIDEA acronym specification on line 80.

Comment 7: Line 81: the performance of the models should be reported. In addition, the rationale of the current study reported in this manuscript should be given. In other words, why do we need other prediction models? Additionally, how their models help physicians take preventive measures while treating patients with 84 COVID. COVID should be replaced with COVID-19.

Answer: Many thanks for the reviewer's comments. Therefore, between lines 82 and 91 we have rewritten and specified the observations described by the reviewer, specifying that the authors of the cited article require predictive models and how such models help clinicians. We appreciate you mentioning these points in your review. Regarding the study mentioned by the reviewer, the performance of the models reported by the study was obtained by hyperparameter optimization. Because of this, they obtained above (90%) accuracy.

Finally, these predictions are useful if the physicians have doubts about a patient with specific symptoms or comorbidities.

Comment 8: Line 88: models predicting hemodialysis is out of the scope of this manuscript and should be removed.

Answer: Thank you very much for the comment. Regarding this related work, we believe this study should be considered because it addresses the problem of using machine learning techniques to detect COVID-19 patients. This study caught our attention because it analyzes data from hemodialysis patients using machine learning techniques. This opens up other research opportunities that may be interesting to analyze with the Epivigila data.

Comment 9: line 155: "in order to know whether these data are 156 relevant or not" is not clear

Answer: Thank you very much for your comment. We decided not to eliminate the data with missing values because we consider that people not reporting their comorbidities can also be used. Identifying the relevant ones considering those patterns can add important information to the study.

Comment 10: Parameters in formulas should be specified.

Answer: Thank you very much for your comment. We have added the parameters of the formulas to improve the manuscript's understanding.

Comment 11: Table 2: p-values were wrong.

Answer: Thanks for the observation. We realize that we misspelled that column in Table 2. We have successfully updated the table.

Comment 12: 4.1. The authors should clearly report in an algorithm and numbers and include patients in their final dataset.

Answer: Thank you very much for the comment. We appreciate the reviewer's comment regarding section 4.1. Regarding this, it is important to point out that we only use descriptive statistics in this section. We do not use algorithms to describe statistics. Unfortunately, we cannot include patient information in the manuscript since we have signed a confidentiality agreement to protect sensitive patient data.

Comment 13: The authors should not only report frequency among positive cases (seemingly Figure 4 and 5) but also the relative frequency (RR or OR) among positive and negative cases. The feature importance ranking should also be provided to interpret the models.

Answer: Thank you very much for the comment. We appreciate what the reviewer suggests to us. We have discussed this and have decided to take the reviewer's suggestion into a second study. All charts described in the article correspond to rankings. However, the metrics suggested by the reviewer require the accompaniment of clinical professionals to help us interpret the results of relative frequency, that is, to make sense of the results. Since this metric shows how often something happens in a specific setting, the results must be interpreted by people working on epidemiological surveillance. Due to the above, we believe that the reviewer's suggestion can be addressed in another paper, which motivates us for future work.

Comment 14: It is not clear how the authors tune their models. They should specify whether they used the testing dataset to fine-tune their models.

Answer: Thank you very much for the comment. The reviewer is absolutely right about how we fine-tuned the machine learning techniques for our study. That said, we have added more information about the parameters and variables used in each technique. With this additional information, we can further specify how we use our models in the manuscript.

Reviewer 2 Report

data collection was not sufficient

Literature survey need to be fine tunned

need more explanation about svm techniques and its parameter implementation.

Each technique was trained on 80% of the available data and tested on 20% why?

In figures x axis and y axis were not mentioned.

Specificity and sensitivity was not discussed in the study.

need more information about physical setup fixed by the authors for collecting the data .

Author Response

Comment 1: data collection was not sufficient

Answer: Thank you very much for your comment. From the complete dataset (7,000,000 patients), 10% corresponds to confirmed cases (700,000) in total. Because of this, we had a unbalanced dataset, and we use probabilistic sampling to obtain the two classes (1, for confirmed; 0, for not confirmed (using suspected and discard). 

Comment 2: Literature survey need to be fine tunned

Answer: Thank you very much for the reviewer's comment. Regarding this comment, in the related works section, we focus on articles close to our study's objectives. The purpose of the section is to show the reader that machine learning techniques are atractive in the clinical community in order to process high volumes of data. The papers we have selected were obtained through a search process whose search string in databases (e.g. WoS, Scopus, Elsevier, etc.) have keywords that match the topics we are addressing in the manuscript. It is expected that the selected articles will differ slightly from the proposal we are addressing. However, the selected articles use machine learning techniques with COVID-19 data for different purposes, which we consider significant for our manuscript. In summary, the related papers address the same objectives as we do, i.e., the analysis of clinical data with machine learning techniques. However, we use real data obtained in real-time through an epidemiological surveillance system used by the Chilean government. This real context brings us an additional challenge to using machine learning techniques.

Comment 3: need more explanation about svm techniques and its parameter implementation.

Answer: Thank you very much for your comment. We have added the parameters of the formulas to improve the manuscript's understanding.

Comment 4: Each technique was trained on 80% of the available data and tested on 20% why?

Answer: Thank you very much for your comment. In the literature (for example, in [1]) it is suggested that the 80/20 ratio between the training set and the test set can be used. On the other hand, due to the bias/variance tradeoff, the training set must not be big because of overfitting.

[1]: Hemdan, E. E. D., Shouman, M. A., & Karar, M. E. (2020). Covidx-net: A framework of deep learning classifiers to diagnose covid-19 in x-ray images. arXiv preprint arXiv:2003.11055.

Comment 5: In figures x axis and y axis were not mentioned.

Answer: Many thanks to the reviewer. We have added the information for each axis in Figure 3.

Comment 6: Specificity and sensitivity was not discussed in the study.

Answer: Thank you very much for your comment. We thank the reviewer for the comment on Specificity and Sensitivity. Regarding Sensitivity, this term is another way of referring to Recall, so we did not consider it. On the other hand, we add Specificity to our analysis because it is relevant for the study of results. Therefore, we add the formula (12), which describes Specificity.

Comment 7: need more information about physical setup fixed by the authors for collecting the data.

Answer: Thank you for your comment. The reviewer is right to ask for more details on how the data is collected. In this regard, in section 3.1, we detail how Epivigila collects data from the whole country from start to finish (report delivered to the Chilean population). In this section, Figure 2 describes the clinical process for collecting patient data. This process is strongly linked to the tasks associated with each actor in the Epivigila system.

Reviewer 3 Report

The authors present a paper entitled "Symptoms and Comorbidities Detection in COVID-19 Patients Using Machine Learning Techniques: A Nationwide Chilean Study ". Despite the relevance of the topic, please address the following concerns:

Minor concerns:

  • Do not use acronyms without specifying their meaning.
  • Please define all the variables in equations 1, 4, and 5.

Author Response

Comment 1: Do not use acronyms without specifying their meaning.

Answer: Thank you very much for your comments. We have indeed used acronyms that we did not specify in the manuscript. Therefore, we have revised the manuscript to detect those acronyms we did not specify and describe them. Additionally, those acronyms that are relevant to the manuscript have been added in the Abbreviations section of the article (see line 350).

Comment 2: Please define all the variables in equations 1, 4, and 5.

Answer: Thank you very much for your comments. We have followed the reviewer's suggestion and detailed the variables used in the equations mentioned by the reviewer.

Reviewer 4 Report

I appreciate the effort of the authors to use the ML algorithms to decipher meanings out of routine healthcare data.  However I feel that the results section need a bit more organization and careful copy editing. For example,

1)line 224:233(4.Results, 4.1 Statistical Analysis), till then you were using ',' as the separator for numbers, but in this paragraph, you use '.' as the number separator, which might cause confusion to the readers.  Also the very first sentence of this paragraph is not a bit confusing. Are you taking about the mean age of males or the whole population?  Better to reword the sentence to avoid the confusion. 

2)I feel that the table 2 can be avoided, anyhow it heading is misleading (Male demographics??)  The table does not add much to the narration in text.

3)Table 4 Better to limit the decimal number of the figures to three decimals; three decimals itself would be difficult to comprehend

4)Table 5  Consider other options to display the results, as it is difficult to comprehend these many decimal values in this complex table. An array or colour matrix could be explored.

5) Reference 5 (line 354-57) is incomplete, it lacks the year of publication.

Author Response

Comment 1: 1)line 224:233(4.Results, 4.1 Statistical Analysis), till then you were using ',' as the separator for numbers, but in this paragraph, you use '.' as the number separator, which might cause confusion to the readers.  Also the very first sentence of this paragraph is not a bit confusing. Are you taking about the mean age of males or the whole population?  Better to reword the sentence to avoid the confusion.

Answer: Many thanks to the reviewer for the comment. Regarding the use of "," and ".", you are absolutely right; we did not realize that we put the numbers in the decimal system of our country. Therefore, we have corrected this and updated the numerical values. On the other hand, I have followed the reviewer's suggestion and we have rewritten the paragraph for better understanding. Therefore, the paragraph starting on line 227 and ending 236 has been updated.

Comment 2: 2)I feel that the table 2 can be avoided, anyhow it heading is misleading (Male demographics??)  The table does not add much to the narration in text.

Answer: Thank you very much for your comment. Table 2 summarizes the demographic data described in section 4.1. On the other hand, the table did not contain information on the female gender, so we have added the missing information. In summary, after a group discussion among the authors of the article, we have decided to keep Table 2 in order to give more support to the argument of the paragraph starting on line 227 and ending on line 236.

Comment 3: 3)Table 4 Better to limit the decimal number of the figures to three decimals; three decimals itself would be difficult to comprehend

Answer: Thank you very much for the comment. We have followed the reviewer's suggestion and rewritten the values to three decimal places. Additionally, we have highlighted the highest values in Table 4 in red.

Comment 4: 4)Table 5  Consider other options to display the results, as it is difficult to comprehend these many decimal values in this complex table. An array or colour matrix could be explored.

Answer: Thank you very much for your comment. Thank you very much for your comment. We have modified the table with 4 decimals to make it easier to read. Additionally, we highlighted the highest values in red color and changed the table to vertical position.

Comment 5: 5) Reference 5 (line 354-57) is incomplete, it lacks the year of publication.

Answer: Many thanks for your comment. We have added the year in the reference.

Round 2

Reviewer 1 Report

Comment 2: Abstract: the "cases" should be specified.

Answer: Thank you very much for your comment. We agree with the reviewer that the word "cases" is ambiguous. Therefore, we rewrote (line 8) the sentence where we specified that the cases are COVID-positive patients.

 Further comment: Now, the authors revised it as "From the group of patients with 9 COVID-19, we selected a sample of 10% confirmed patients to execute and evaluate the techniques". How about the control group? Similarly, the confused expression in result 4.1 should be revised. The subtitle should be rephrased as well. 

Comment 3: Why selected only 10% of confirmed cases?

Answer: Thank you very much for your comment. In the original dataset, the confirmed cases are only the 10\% of the total. The other classes (90\%) correspond to discard cases or suspected cases, so we tried to maintain the proportion in each sample.

Further comment: please revise accordingly; it is still confusing now.

Comment 4: SVM performed better by how much difference? Need to give a detailed quantitative description of the comparison

Answer: Thank you very much for the comment. Undoubtedly, the reviewer addresses an interesting aspect of our article. In our study, we have used performance metrics that are defined in equations 10, 11, 12, 13, and 14. Based on these metrics, in section 4.2, we discuss the results of each machine learning technique. In this regard, we realized that SVM had the best recall in all age ranges. Although the literature, in general, suggests that better precision determines the performance of a machine learning technique, this judgment depends on the context. Given that our study analyzes the predictions of the outputs of machine learning techniques, the recall should be high since we are also interested in false negatives. Having said this, SVM presented a high recall in all age ranges, so we consider that SVM had a better performance than the rest of the techniques.

Further comment: please revise the part in the abstract. Evidence should be given to inform the readers how SVM is performing better than other models. 

Comment 5: Features used in the models should be reported.

Answer: Thank you very much for your comment. For every model, the features used were the 14 symptoms and the 11 comorbidities (see Table 1). Additionally, the age of the patient was used. Finally, the classes were divided into confirmed and discarded.

Further comment: it is not clear what discarded meant. 

Comment 7: Line 81: the performance of the models should be reported. In addition, the rationale of the current study reported in this manuscript should be given. In other words, why do we need other prediction models? Additionally, how their models help physicians take preventive measures while treating patients with 84 COVID. COVID should be replaced with COVID-19.

Answer: Many thanks for the reviewer's comments. Therefore, between lines 82 and 91 we have rewritten and specified the observations described by the reviewer, specifying that the authors of the cited article require predictive models and how such models help clinicians. We appreciate you mentioning these points in your review. Regarding the study mentioned by the reviewer, the performance of the models reported by the study was obtained by hyperparameter optimization. Because of this, they obtained above (90%) accuracy.

Finally, these predictions are useful if the physicians have doubts about a patient with specific symptoms or comorbidities.

Further comment: since Awal and many other researchers might have reported models to predict COVID-19, the authors should review their limitations, report the gap in knowledge for the rationale of this study, and preferably compare the performance of other models simultaneously in their study. 

Comment 8: Line 88: models predicting hemodialysis is out of the scope of this manuscript and should be removed.

Answer: Thank you very much for the comment. Regarding this related work, we believe this study should be considered because it addresses the problem of using machine learning techniques to detect COVID-19 patients. This study caught our attention because it analyzes data from hemodialysis patients using machine learning techniques. This opens up other research opportunities that may be interesting to analyze with the Epivigila data.

Further comment: The model proposed by Monahan et al. is to predict hemodialysis with undetected COVID-19 infection. The authors did not describe any machine learning model in this paragraph. This paragraph is still redundant. 

Comment 9: line 155: "in order to know whether these data are 156 relevant or not" is not clear

Answer: Thank you very much for your comment. We decided not to eliminate the data with missing values because we consider that people not reporting their comorbidities can also be used. Identifying the relevant ones considering those patterns can add important information to the study.

Further comment: rewording is needed.  

Comment 12: 4.1. The authors should clearly report in an algorithm and numbers and include patients in their final dataset.

Answer: Thank you very much for the comment. We appreciate the reviewer's comment regarding section 4.1. Regarding this, it is important to point out that we only use descriptive statistics in this section. We do not use algorithms to describe statistics. Unfortunately, we cannot include patient information in the manuscript since we have signed a confidentiality agreement to protect sensitive patient data.

Further comment: same further comment provided in comment 2. 

Comment 13: The authors should not only report frequency among positive cases (seemingly Figure 4 and 5) but also the relative frequency (RR or OR) among positive and negative cases. The feature importance ranking should also be provided to interpret the models.

Answer: Thank you very much for the comment. We appreciate what the reviewer suggests to us. We have discussed this and have decided to take the reviewer's suggestion into a second study. All charts described in the article correspond to rankings. However, the metrics suggested by the reviewer require the accompaniment of clinical professionals to help us interpret the results of relative frequency, that is, to make sense of the results. Since this metric shows how often something happens in a specific setting, the results must be interpreted by people working on epidemiological surveillance. Due to the above, we believe that the reviewer's suggestion can be addressed in another paper, which motivates us for future work.

Further comment: the current presentation is still misleading and biased. 

Comment 14: It is not clear how the authors tune their models. They should specify whether they used the testing dataset to fine-tune their models.

Answer: Thank you very much for the comment. The reviewer is absolutely right about how we fine-tuned the machine learning techniques for our study. That said, we have added more information about the parameters and variables used in each technique. With this additional information, we can further specify how we use our models in the manuscript.

Further comment: the revision cannot be found in the current manuscript. 

Author Response

1) Further comment: Now, the authors revised it as "From the group of patients with 9 COVID-19, we selected a sample of 10% confirmed patients to execute and evaluate the techniques". How about the control group? Similarly, the confused expression in result 4.1 should be revised. The subtitle should be rephrased as well.

Answer: Thank you very much for your comment. In order to explain in more detail everything related to the control group, we have added a new paragraph (first paragraph, lines 151 to 161) in section 3.2. In this paragraph, we describe the quantitative details of the control group. Regarding section 4.1, for better understanding, we have changed the name of the section and added more qualitative information (e.g. Figures 6 and 7) following the reviewer's suggestion.

2) Further comment: please revise accordingly; it is still confusing now.

 Answer: Thank you very much for your comment. In the first paragraph of section 3.2 (lines 151 to 161) we have detailed why we selected 90% and 10% of the total data. We focused on quantitatively explaining the values obtained in both data sets. We also explain the concepts of confirmed, suspected and discarded cases.

3) Further comment: please revise the part in the abstract. Evidence should be given to inform the readers how SVM is performing better than other models.

Answer: Thank you very much for your comment. We have followed the reviewer's suggestion and detailed in the abstract why SVM is better than the other techniques. In lines 10, 11 and 12 we describe that SVM is better based on recall, accuracy, f1 score and the AUC curve.

4) Further comment: it is not clear what discarded meant.

Answer: Thank you very much for your comment. We have described in the first paragraph of section 3.2 the definition of confirmed, suspected and discarded patients.

5) Further comment: since Awal and many other researchers might have reported models to predict COVID-19, the authors should review their limitations, report the gap in knowledge for the rationale of this study, and preferably compare the performance of other models simultaneously in their study.

Answer: Thank you very much for your comment. There is no doubt that what the reviewer mentions is very relevant. Comparing the different models presented in the literature is an interesting study to conduct. Since the scope of our study is to use machine learning techniques on a little or almost unexplored dataset, our first goal is to report the results using this dataset (Epivigila) with known machine learning techniques and discuss the corresponding results. However, following the reviewer's suggestion to compare other models using the Epivigila dataset is very motivating. We will undoubtedly find interesting results using this dataset with real data, which merits another paper.

6) Further comment: The model proposed by Monahan et al. is to predict hemodialysis with undetected COVID-19 infection. The authors did not describe any machine learning model in this paragraph. This paragraph is still redundant.

 Answer: Thank you very much for your comment. Indeed, in the previous manuscript version, we did not explicitly describe which machine learning technique the authors of the cited paper used. Therefore, in lines 94, 95 and 96 we explicitly describe the machine learning technique used by the authors.

7) Further comment: rewording is needed. 

 Answer: Thank you for your comment. In order to better explain our concept of missing values, we have described between lines 172 and 179 how the vector describing the occurrences of symptoms and comorbidities is generated. In that explanation, we describe the interpretation of missing values.

8) Further comment: same further comment provided in comment 2.

Answer: Thank you very much for your comment. Regarding comment 2, in order to better explain the descriptive statistics in section 4.1, we have detailed in the first paragraph of the section demographic data relevant to the understanding of the manuscript. For example, (i) in the first paragraph of section 4.1, we mention interquartile range results for both males and females (with much more detail in Table 2); (ii) in Figure 3 we show the distribution of confirmed COVID-19 patients by age; (iii) and the frequency of symptoms and comorbidities detected in COVID-19 positive patients. On the other hand, the reviewer asked us to include patients in the final dataset in the initial comment. Although we strongly agree that following the reviewer's suggestion is an excellent idea, patient privacy restrictions do not allow us to add patient information to the manuscript. Finally, as mentioned in the previous answer, we have used known formulas to describe section 4.1 and have not used any specific algorithm. We hope that with this explanation, the reviewer can be satisfied with the observation he/she made.

9) Further comment: the current presentation is still misleading and biased.

Answer: Thanks to the reviewer for the comment. After analyzing the results described in section 4.1, we have followed the reviewer's suggestion. Therefore, Figures 6 and 7 depict the relative frequency percentage of confirmed and discarded patients. With this additional information, we mitigate the bias of the results described in section 4.1.

10) Further comment: the revision cannot be found in the current manuscript.

Answer: Thank you very much for your comment. Information regarding the parameters of each machine learning technique can be found in lines 192-195, 200-201, 212, 213-215, and 216-218.